# Anti-Cancer Effects of Green Tea Polyphenols Against Prostate Cancer

**DOI:** 10.3390/molecules24010193

**Published:** 2019-01-07

**Authors:** Yasuyoshi Miyata, Yohei Shida, Tomoaki Hakariya, Hideki Sakai

**Affiliations:** Department of Urology, Nagasaki University Graduate School of Biomedical Sciences, Nagasaki 852-8501, Japan; yshida.urodr@gmail.com (Y.S.); erbb2jp@yahoo.co.jp (T.H.); hsakai@nagasaki-u.ac.jp (H.S.)

**Keywords:** green tea, prostate cancer, anti-cancer effects, molecular mechanism

## Abstract

Prostate cancer is the most common cancer among men. Green tea consumption is reported to play an important role in the prevention of carcinogenesis in many types of malignancies, including prostate cancer; however, epidemiological studies show conflicting results regarding these anti-cancer effects. In recent years, in addition to prevention, many investigators have shown the efficacy and safety of green tea polyphenols and combination therapies with green tea extracts and anti-cancer agents in in vivo and in vitro studies. Furthermore, numerous studies have revealed the molecular mechanisms of the anti-cancer effects of green tea extracts. We believe that improved understanding of the detailed pathological roles at the molecular level is important to evaluate the prevention and treatment of prostate cancer. Therefore, in this review, we present current knowledge regarding the anti-cancer effects of green tea extracts in the prevention and treatment of prostate cancer, with a particular focus on the molecular mechanisms of action, such as influencing tumor growth, apoptosis, androgen receptor signaling, cell cycle, and various malignant behaviors. Finally, the future direction for the use of green tea extracts as treatment strategies in patients with prostate cancer is introduced.

## 1. Introduction

Prostate cancer (PC) is the most common cancer in men worldwide. In addition, it is one of the major causes of cancer-related mortality. Although the detailed pathogenesis and carcinogenic processes are not completely understood, the relationships between tumor microenvironment and cancer cells might play important roles in cancer-related processes [1,2]. The frequency and mortality are known to depend on the place of residence and race [3,4]. Conversely, dietary habits have been shown to affect the risk and development of PC. In fact, many epidemiological studies have shown that dietary habits, including green tea consumption, significantly decrease the risk of carcinogenesis and development of PC [5,6,7,8].

Green tea, extracted from the leaves of *Camellia sinensis* (Theaceae family), has been widely consumed as a beverage in Asian countries such as China, Japan, Korea, and India for centuries [4,9,10,11,12]. Green tea catechins (GTCs) are a type of green tea polyphenols (GTP) that are present at high levels in green tea, and are the source of its distinctive bitter taste. GTCs present in green tea include (−)-epigallocatechin-3-gallate (EGCG); (−)-epicatechin (EC); (−)-epigallocatechin (EGC); and (−)-epicatechin-3-gallate (ECG) [13]. Among these GTCs, in vitro and animal studies have shown that EGCG is highly bioactive and targets the molecular pathways implicated in prostate carcinogenesis [7,11,12,14,15].

In general, the growth of hormone-naïve PC cells is strongly suppressed by androgen deprivation. In addition, the prognosis of patients with organ-confined PC is good with radical prostatectomy and radiotherapy. Therefore, in these patients, there is little need for treatments involving the use of green tea or GTPs. Hormonal therapy, including androgen deprivation therapy, is recognized as the standard for these patients even in the case of advanced or metastatic disease. However, unfortunately, most patients develop castration-resistant prostate cancer (CRPC) despite therapeutic suppression of testosterone levels. In addition, the prognosis of CRPC patients is poor owing to the high malignant potential and aggressiveness of CRPC. CRPC is considered to involve numerous gene mutations and alternate signaling pathways. Therefore, treatment strategies targeting a few pathways are not effective, leading to the rapid development of chemoresistance. Thus, the development of new treatment strategies is essential to improve the prognosis of CRPC patients.

PC has a long latency period and is typically diagnosed in elderly men. Therefore, chemoprevention strategies have been studied in detail by many investigators [16,17]. Conversely, safety and cost are important since long-term periodic administration is necessary for the chemoprevention of PC. In addition, an ideal agent for the chemoprevention of PC would also prevent other diseases and promote the maintenance of healthy conditions. Thus, natural compounds, including green tea, rather than chemical agents, are the major subjects of in vivo, in vitro, and epidemiological studies on the chemoprevention of PC [18]. In this review, we paid special attention to three aspects of the effects of green tea on PC: the chemopreventive effect against PC, therapeutic effects for treating PC, and the molecular mechanisms of such anti-cancer effects. Several prospective trials are investigating the chemoprevention of PC by green tea. Further, basic research is being conducted with regard to the therapeutic effect of green tea against PC. Recently, some studies have suggested the preoperative administration of green tea before radical prostatectomy. Therefore, interest in the therapeutic effects of green tea is increasing. However, the limitations of the anti-cancer effects and the clinical usefulness of green tea must also be understood to evaluate the prevention and treatment strategies by using green tea-based methods. Herein, we present data on green tea with respect to PC and believe that these data will be useful for future researchers.

## 2. Anti-Cancer Effects of Green Tea

### 2.1. Case-Control Studies

Several case-control studies have investigated the preventive effects of green tea for PC. For example, a case-control study with 140 PC cases and an equal number of hospital patients as controls was performed in Japan [19]. This study showed an inverse correlation between green tea consumption and PC risk, although it did not reach the level of significance [19]. Conversely, another case-control study in China showed that increasing the frequency, duration, and quantity of green tea consumption could lead to a lower risk of PC [7]. In this study, a hospital-based 1:2 case-control design (130 cases and 274 hospital controls) was used to investigate the association between green tea consumption and PC. This was the first study providing comprehensive evidence of the protective effect of green tea against PC. The adjusted odds ratios (OR) compared with those of men who never or seldom drank green tea were 0.28 (95% CI: 0.17, 0.47) for those drinking tea, 0.12 (95% CI: 0.06, 0.26) for those drinking tea for over 40 years, and 0.27 (95% CI: 0.15, 0.48) for those drinking more than 3 cups (1 L) per day. These results suggest that green tea has a protective effect against PC. A second case-control study in China, which consisted of 250 PC patients and 500 controls, also showed that green tea consumption was associated with a decreased risk of PC (OR = 0.59, 95% CI: 0.40, 0.87) [20]. A summary of these studies is shown in Table 1.

### 2.2. Prospective Studies

#### 2.2.1. Green Tea Consumption

In a previous prospective study, Severson et al. [21] showed an association between several diets and the development of PC in a cohort of 7999 Japanese men living in Hawaii. They concluded that subjects who consumed prominently oriental food items, including green tea, had a decreased risk of PC, but the trend was not statistically significant. Similarly, Allen et al. [22] investigated 193 patients among 18,115 Japanese men living in Hiroshima and Nagasaki (Life Span Study Cohort) to analyze the relationship between green tea intake and PC; they showed that men who drank green tea 5 or more times per day had a 29% increased risk of PC compared with those who drank green tea less than once per day. However, this was not statistically significant. Such negative correlations were also reported in several other studies. In short, one prospective cohort study of 110 cases among 19,561 men living in 14 municipalities of Miyagi Prefecture in Japan showed that the multivariate hazard ratios (HRs) for PC associated with drinking 1–2, 3–4, and 5 or more cups of green tea per day, as compared with consuming less than one cup per day, were 0.77 (95% CI: 0.42, 1.40), 1.15 (95% CI: 0.69, 1.94), and 0.85 (95% CI: 0.50, 1.43), respectively (*p* = 0.81) [23].

In contrast to these studies, a prospective cohort study consisting of 114 patients among 49,920 Japanese men showed that green tea consumption was associated with a decreased risk of advanced PC [23]. Furthermore, they detected a dose-dependent inverse relation for the risk of advanced PC (*p* = 0.01), and the risk in men who consumed ≥5 cups of green tea/day was lower (relative risk: 0.52, 95% CI: 0.28, 0.96) than in those who drank <1 cup/day. However, this study found no significant relationship between green tea consumption and the risk of localized PC [24]. From these findings, green tea can be speculated to suppress the progression from localized to advanced disease in PC. This was the first prospective study to investigate the association between green tea consumption and PC at distinct stages of progression and to identify the preventive effects of green tea on advanced PC [24]. In addition to studies in Japanese men, another prospective cohort of Chinese men in Singapore showed an HR with a 95% CI of 0.79 to 1.47 [25]. A summary of these prospective cohort studies is shown in Table 2.

#### 2.2.2. Green Tea Catechin Intake

Regarding GTCs and PC risk, three randomized clinical trials have been reported, two of which suggested the efficacy of GTCs in the prevention of PC. Bettuzzi et al. [5] reported the safety and efficacy of GTCs for the chemoprevention of PC in high-grade prostate intraepithelial neoplasia (HG-PIN) volunteers. In this double-blind, placebo-controlled study, 60 Caucasian male volunteers bearing HG-PIN lesions to whom no other therapy was administered were enrolled to investigate whether the administration of GTCs (EGC, 5.5%; EC, 12.24%; EGCG, 51.88%; ECG, 6.12%; total GTCs, 75.7%; caffeine, <1%) could prevent malignancy. The GTC-treated men took three GTC capsules of 200 mg each (total 600 mg/day). After 1 year, only one tumor was diagnosed among the 30 GTC-treated men (incidence rate, 3%), whereas nine tumors were found among the 30 placebo-treated men (incidence rate, 30%). This was the first study showing that GTCs are safe and effective for treating premalignant lesions before the development of PC. Moreover, the administration of GTCs reduced lower urinary tract symptoms. These results raised the possibility that GTCs could be useful to prevent the development of PC and for the treatment of the symptoms of benign prostate hyperplasia. Two years later, a follow-up study was reported by Brausi et al. [26]. The mean follow-up from the end of GTC administration was 23.3 months for the placebo arm and 19.1 months for the GTC arm of the study. Kaplan–Meier analysis of the final results showed that GTC treatment led to an almost 80% reduction in PC diagnosis, from 53 to 11%. The final difference in cancer prevalence was highly significant (*p* < 0.01) by chi-squared test analysis. These results suggested that the inhibition of PC progression by GTCs was long lasting.

### 2.3. Meta-Analyses

A meta-analysis indicated that the consumption of green tea might have a protective effect on PC in Asian populations, especially in Chinese populations [27]. The OR of PC indicated a borderline significant association in Asian populations at the highest level of green tea consumption compared to the no/lowest level of consumption (OR = 0.62, 95% CI: 0.38, 1.01). According to this study, black tea consumption did not show a similar protective effect on PC. The authors stated that further prospective cohort studies are needed to obtain a definitive conclusion regarding the protective effect of green tea on PC. Guo et al. [28] also reported a meta-analysis of green tea consumption and PC incidence. Cohort or case–control studies and randomized controlled trials (RCTs) were included and analyzed accordingly. The result of dose–response meta-analysis revealed that each cup/day increase in green tea intake decreased the risk of PC with a RR of 0.954 (95% CI: 0.903, 1.009) for all studies, 0.989 (95% CI: 0.957, 1.023) for cohort studies, and 0.893 (95% CI: 0.796, 1.002) for case–control studies. These studies showed that higher green tea consumption (more than 7 cups/day) significantly decreased PC risk. Conversely, another meta-analysis performed by Lin et al. [29] found that green tea consumption in the Asian population did not reduce PC risk (OR = 0.82, 95% CI: 0.42, 1.21), although green tea is generally considered to be rich in strongly anti-cancer phytochemicals. Thus, meta-analyses showed no general agreement on the preventive effects of green tea for PC.

## 3. Anti-Cancer Effects of Green Tea at the Molecular Level

### 3.1. Tumor Growth

#### 3.1.1. In Vitro Studies

As mentioned above, EGCG is recognized to have the strongest anti-cancer activity among the GTPs. Therefore, first, we introduce the relationship between EGCG and tumor growth in PC. By using in vitro studies, many investigators have shown that EGCG suppressed tumor growth of androgen-sensitive (LNCaP and CWR22Rv1) and androgen-independent PC cells (DU-145 and PC-3 cells) in a dose-dependent manner [30,31,32,33,34,35]. In addition, another study showed that EGCG suppressed cell proliferation of androgen-sensitive LNCaP 104-S, androgen-independent LNCaP 104-R1, and androgen-adapted LNCaP R1Ad cells in both the presence and absence of androgen [36]. Furthermore, EGCG inhibited the growth of hormone-refractory C4-2 cells that were able to grow in hormone-free medium [35].

Data on the relationship between cell proliferation and EGC, ECG, or EC in PC in vitro are limited. However, EGCG, ECG, and EGC, but not EC, have been reported to have significant inhibitory effects on cell proliferation in DU145 cells [34]. Interestingly, the same study also showed that the suppressive effect was in the order ECG > EGCG > EGC [34]. Thus, a general agreement is that EGCG can inhibit the growth of PC cells regardless of androgen-dependency in vitro. More detailed studies are necessary to determine the anti-proliferative effect of other components of GTPs.

#### 3.1.2. In Vivo Studies

Regarding preventive effects of GTP, there is a report that oral infusion of 0.1% GTP (wt/vol) fluid (EGCG 62%, ECG 24%, EGC 5%, EC 6%, and caffeine ≈ 1%) from 8 to 32 weeks of age showed anti-carcinogenic activity, including suppression of cancer cell proliferation, in the autochthonous transgenic adenocarcinoma of the mouse prostate (TRAMP) model [37].

On the other hand, in athymic nude mice, intra-peritoneal infusion of EGCG suppressed tumor growth and reduced the size of AR-negative androgen-insensitive PC-3 xenografts and androgen-positive androgen-independent LNCaP 104-R1 xenografts, whereas EC, EGC, or ECG did not [38]. In addition, Chuu et al. [36] reported that intra-peritoneal injection (1 mg/day for 11 weeks) of EGCG caused a 40% reduction in the tumor volume of relapsed tumors. Interestingly, their results also showed that serum prostate-specific antigen (PSA) levels and PSA density (serum PSA level divided by tumor volume) values in the EGCG-treated mice were significantly lower than those in control mice [36]. Based on these data, the authors suggested that EGCG could inhibit AR signaling and PSA promoter activity in relapsed tumors. Thus, intra-peritoneal infusion of EGCG might have an inhibitory effect on tumor growth in both androgen-sensitive and androgen-independent PC. However, intra-peritoneal infusion is unrealistic in a clinical setting for PC patients.

Regarding the anti-cancer effects of administering GTP orally, the intake of 0.1% GTP fluid significantly suppressed the growth of implanted tumors in athymic nude mice implanted with androgen-sensitive human PC cells (CWR22Rv1) [39]. This study also showed that serum PSA levels in mice treated with GTP (4.95 ± 1.23 ng/mL) were significantly (*p* < 0.01) lower than those in control mice (13.95 ± 0.82 ng/mL). Similar results on decreasing the serum PSA level by EGCG were obtained by another in vivo study [32]. Conversely, some authors have suggested that EGCG can suppress early-stage, but not late-stage, PC in TRAMP mice [40]. Briefly, in TRAMP mice orally administered 0.06% EGCG in tap water, the frequency of HG-PIN was reduced from 100% to 17% at 12 weeks of age; however, such a significant effect was not observed at 28 weeks of age. Similarly, downgrading of tumors following EGCG administration was detected at 14 weeks of age, but not at 28 weeks of age. Thus, similar to the findings of in vitro studies, EGCG suppressed tumor growth of both androgen-sensitive and androgen-independent PC, even when administered orally. Conversely, oral administration of 0.07% GTP solution, containing EGCG 388 ± 12, EGC 204 ± 4, ECG 64 ± 7, EC 44 ± 2, and catechin 7 ± 1 mg/L, increased the number of Ki-67-positive PC cells in the prostate tissues of a xenograft model injected with androgen-sensitive LAPC-4 PC cells [41]. However, the content of GTCs used in in vivo studies should be considered for a better understanding of the effects of GTCs. In short, as shown in this section, the concentrations and compositions of the GTCs used differed in the various studies, and these differences may affect their biological effects.

### 3.2. Cell Death

#### 3.2.1. Apoptosis—In Vitro Studies

Many studies have shown that EGCG treatment resulted in the induction of apoptosis in both LNCaP and DU-145 cells [30,33,42,43]. However, several molecules associated with apoptosis are reported to be affected by EGCG in in vitro studies. For example, EGCG stimulates apoptosis in LNCaP cells by decreasing the levels of anti-apoptotic molecules such as B-cell lymphoma-2 (Bcl-2) and increasing those of pro-apoptotic ones such as Bax [44]. Conversely, one study found that EGCG, EGC, and ECG, but not EC, induced apoptosis in DU-145 cells [45]. Similarly, other investigators have shown that EGCG treatment reduced the levels of anti-apoptotic molecules such as phosphatidylinositol-3-kinase (PI3K) and phospho-Akt and increased the activity of pro-apoptotic factors, including extracellular signal-regulated protein kinase (Erk) 1/2 in both androgen-sensitive and androgen-independent PC cells (LNCaP and DU-145 cells) [45,46]. Conversely, apoptosis induced by GTCs, including EGCG, EGC, ECG, and EC, has been reported not to be related to the members of the Bcl-2 family, as EGCG did not alter the expression of anti-apoptotic molecules, including Bcl-2 and Bcl-xL or the pro-apoptotic Bad in DU145 cells [34]. This suggests that EGCG can induce apoptosis in both androgen-sensitive and androgen-independent PC cells via the regulation of various factors; however, further studies are necessary to reveal whether EGC and ECG can affect the apoptotic process in PC cells. Conversely, EC has been speculated to play a minimal role in cell survival and cell death in these cells.

#### 3.2.2. Apoptosis—In Vivo Studies

Several investigators have shown that GTP induced apoptosis in PC tissues of PC animal models. For example, the oral administration of GTP, which contains four major polyphenolic constituents, i.e., EGCG (62%), EGC (24%), EC (6%), and ECG (5%), for 24 weeks significantly increased apoptosis in the prostate tissues of TRAMP mice [37]. Similar pro-apoptotic activity of oral intake of the same GTP was confirmed in TRAMP mice by other investigators [47]. Conversely, the oral intake of 0.3% GTCs (EGC 5.5%, EC 12.2%, EGCG 51.9%, ECG 6.1%; total GTCs 75.7%, caffeine <1.0%) in drinking water suppressed carcinogenesis in TRAMP mice via the regulation of the apoptosis-related molecule clusterin [48], which is reported to be associated with oncogenesis and metastasis in PC [49,50]. In addition to TRAMP mice, several xenograft models injected with PC cells, such as androgen-sensitive LNCaP, CWR22Rv1, or LAPC-4 PC cells [32,39,41], and androgen-independent PC-3 cells [32], showed similar pro-apoptotic activity after oral administration of GTP. Oral administration of a 0.06% solution of EGCG is also reported to induce apoptosis in the prostate tissues of TRAMP mice [40]. Thus, in contrast to in vitro studies, many studies have investigated the effects of GTP solution, but not those of EGCG alone, to elucidate the pro-apoptotic activity in animal models.

Many studies have described the molecular mechanisms of the pro-apoptotic activity of GTP in animal models. Generally, the Bcl-2 family is considered to be one of the most important regulators of chemical-induced apoptosis in many types of cancers, including PC [51,52]. Therefore, we reviewed the influence of the oral administration of GTP on the members of the Bcl-2 family in PC animal models. GTP treatment of athymic nude mice implanted with androgen-sensitive human LNCaP and CWR22Rnu1 cells and androgen-independent PC-3 cells resulted in the induction of apoptosis accompanied by a decrease in Bcl-2 and an increase in Bax expression [32,39]. In addition to the xenograft model, TRAMP mice showed similar effects on Bcl-2 and Bax expression in PC tissues [53].

#### 3.2.3. Other Kinds of Cell Death

In addition to apoptosis, other kinds of cell death, such as anoikis, necrosis, autophagy, and entosis, are associated with pathological characteristics and malignant behavior of PC [54,55]. Actually, several studies showed that a variety of GTPs induced such kinds of cell death of PC cells in vitro. For example, polyphenon E, which is a standardized green tea extract, and EGCG were reported to induce PC cell death though anoikis and autophagy in PC cells [56]. On the other hand, various natural compounds including GTPs affect tumorigenesis, tumor progression, and anti-cancer effects by regulating autophagy in PC [57,58]. Thus, GTPs can modulate various types of cell death, and such GTP-induced cell death is speculated to be associated with the malignant aggressiveness of PC. We emphasize that understanding the mechanisms of GTP-induced cell death is important for assessing the usefulness and future challenges of GTP-using strategies.

### 3.3. Cell Cycle

The cell cycle regulates the birth, growth, and death of cells under physiological conditions, and dysregulation of this system is associated with carcinogenesis, tumor growth, and anti-cancer therapy efficacy [59,60]. GTP is known to affect the regulation of the cell cycle and cell cycle-related molecules in many types of malignancies [61]. EGCG treatment in vitro led to G(0)/G(1) phase arrest and dysregulation of the cell cycle in both androgen-sensitive LNCaP cells and androgen-insensitive DU145 cells [30,43]. Conversely, various types of cell signaling molecules and regulators of the cell cycle mediate this EGCG-induced cell cycle regulation [45]. p53 is one of the best-studied cell cycle regulators, and EGCG induced apoptosis via the stabilization of p53 in LNCaP cells [44]. In short, when LNCaP cells were treated with 20, 40, 60, and 80 μM EGCG for 24, 48, and 72 h, the two higher concentrations induced apoptosis after 24 and 48 h of treatment; in addition, apoptosis was significantly induced after 72 h of treatment at all doses [44]. Furthermore, this study also showed such a dose-dependent function of EGCG in the levels of protein expression, transcriptional activation, and p53 stability in LNCaP cells [44]. Conversely, regarding the relationship between safety and EGCG concentration, 100 μM of EGCG was reported to have relatively minor effects on the viability of human non-tumoral prostate cells [62]. Thus, EGCG can induce apoptosis of LNCaP cells at concentrations that are relatively safe for normal cells. Another study also found similar p-53-mediated pro-apoptotic function of EGCG in LNCaP cells [30]. Furthermore, in both LNCaP and DU145 cells, EGCG increased the expression of WAF1/p21, KIP1/p27, INK4a/p16, and INK4c/p18 and decreased the expression of cyclin D1, cyclin E, cyclin-dependent kinase (CDK) 2, CDK4, and CDK6 in a dose- and time-dependent manner [63]. In addition, the same authors also found that EGCG increased the binding of cyclin D1 to WAF1/p21 and KIP1/p27 and decreased the binding of cyclin E to CDK2 [63]. From these results, the authors suggested that EGCG causes an induction of G1 phase cyclin-dependent kinase inhibitors (CDKIs), which inhibit the operation of the cyclin–CDK complexes in the G0/G1 phase of the cell cycle, thereby causing cell cycle arrest. This might be an irreversible process, ultimately leading to apoptotic cell death. Thus, EGCG might affect the regulation of the cell cycle via complex mechanisms, including various types of molecules, in both androgen-sensitive and androgen-independent PC cells. However, EGCG has been reported to lead to increased levels of p53 in LNCaP cells, but not in DU145 cells [30]. Thus, the molecular mechanism affecting the cell cycle is probably dependent on androgen. In fact, in DU-145 cells, EGCG resulted in a significant induction of Cip1/p21, Kip1/p27, and CDKI/CDK binding and a decrease in CDK4; however, a similar significant change was not observed in CDK2, cyclin D1, or cyclin E [46,64]. Finally, we speculate that treatment with EGCG causes cell cycle dysregulation in both androgen-sensitive and androgen-independent human PC cells. On the other hand, unfortunately, there are few reports on in vivo effects of GTPs in the cell cycle. Therefore, more detailed in vivo and in vitro studies are necessary to clarify their molecular mechanisms and the requirement for androgen dependency.

### 3.4. Androgen Receptor

The androgen receptor (AR) plays important roles in both early and advanced stages of PC by regulating tumor growth, apoptosis, and invasion, and the expression of many target genes implicated in these malignant behaviors [65,66,67]. In fact, androgen deprivation therapy is a standard therapy for many PC patients, such as those with metastatic disease.

In LNCaP cells, EGCG downregulated the mRNA and protein expression of AR [68]. In addition, this study showed that Sp1 DNA binding is the target of this EGCG-induced activity [68]. Another study showed that EC, EGC, and EGCG induced cell death in prostate cancer cells and suppressed AR activation and AR-regulated gene transcription by regulating histone acetyl-transferase (HAT) activity in LNCaP cells [69]. They also found that EGCG had the strongest inhibitory effect on the histone acetyl-transferase activity and was able to suppress AR-regulated gene transcription. Conversely, EGCG was observed to reduce the viability of PC-3 cells; however, this effect was not significant. EC and EGC did not affect viability in PC-3 cells [69]. Similarly, HAT activity was not influenced by EC, EGC, or EGCG in PC-3 cells. Thus, GTPs, especially EGCG, suppressed cancer cell growth by modulating the acetylation of AR by HAT activity in androgen-dependent, but not in androgen-independent, PC cells.

In animal models, EGCG administration led to the inhibition of AR expression and nuclear transcription of AR, followed by the suppression of AR-dependent functions [34,36]. In short, PC xenografts of 22Rv1 cells treated with EGCG (1 mg/day, 3 times/week, intra-peritoneal for 6 weeks) showed significantly reduced AR expression compared to that of the vehicle-treated cells [36]. In addition, oral administration of 0.06% EGCG solution led to a decrease in AR expression [40]. Thus, many investigators suggest that EGCG might significantly affect the malignant behavior of PC via the regulation of the AR pathway. However, the effects of other components of GTP are not completely understood in vivo or in vitro.

We also emphasize the importance of understanding the influence of GTPs on the anti-cancer effects of anti-androgen agents and the relationships between GTPs and testosterone or dihydrotestosterone in PC. Actually, in 1995, EGCG and ECG were reported as potent inhibitors of 5α-reductase in a rat model [70]. On the other hand, in an animal model, GTPs, comprised of approximately 27% catechins, 8.0% caffeine and 0.4% theobromine, with EGCG representing ≥50% of the total catechins, are reported to inhibit testosterone production in Leydig cells [71]. In addition, another report showed that intragastric administration of 100 or 50 mg/kg EGCG suppressed testosterone-induced benign prostate hyperplasia [72]. On the other hand, there is a report on dihydrotestosterone sensitizing LNCaP cells to apoptosis induced by EGCG [33]. Interestingly, this study also showed that a sub-apoptotic dose of EGCG (8 μM) switched dihydrotestosterone from functioning as a growth promoter to a growth inhibitor in the same cell line [33]. Such dose-dependent GTP-induced activity modulation was not observed for testosterone. However, no additional investigations were conducted in this regard. Further detailed studies on the relationships between GTPs and AR-related pathways are necessary to elucidate the clinical usefulness of green tea in patients with PC.

### 3.5. Matrix Metalloproteinases

Matrix metalloproteinases (MMPs) play important roles in tumor development through the degradation of the extracellular matrix of tissues surrounding tumors and the regulation of cell survival in PC [73,74,75]. Among the various MMPs, two members of the MMP family, MMP-2 and MMP-9, are recognized as the most important factors for cancer cell invasion and metastasis in many types of cancers, including PC [66,75,76,77].

In one study, fibroblast-conditioned medium stimulated the activities of MMP-2 and MMP-9 in DU-145 cells, although EGCG inhibited such activation, concomitant with a marked inhibition of the phosphorylation of ERK1/2 and p38, but not of JNK [78]. Furthermore, EGCG inhibited androgen-induced expression of the inactive pro-MMP2 in LNCaP cells [78]. From this finding, the authors suggested that EGCG might play important roles in inhibiting not only the synthesis of MMPs but also the conversion of the pro-forms to their active forms. Furthermore, in both DU145 and PC-3 cells, the gelatinase activities of MMP-2 and MMP-9 were suppressed by a nutrient mixture containing lysine, proline, ascorbic acid, and green tea extract [79].

Conversely, the tissue inhibitor of metalloproteinases (TIMPs) is recognized as an inhibitory factor of MMPs in malignancies [80]. The balance between MMP and TIMP activities is recognized as an important determinant of malignant aggressiveness in several cancers [81,82,83]. Unfortunately, data on the influence of GTP on TIMPs in PC are limited. However, one study indicated that a nutrient mixture containing lysine, proline, ascorbic acid, and green tea extract increased the activities of TIMP-1 and TIMP-2 in both DU145 and PC-3 cells [79].

### 3.6. Insulin-Like Growth Factors

One study showed that serum levels of insulin-like growth factor (IGF)-1 and IGF-binding protein (IGFBP-3) and the IGF-1/IGFBP-3 ratio were significantly lower after treatment with polyphenon E, which contained 800 mg of EGCG and lesser amounts of EC, EGC, and ECG (a total of 1.3 g of tea polyphenols) [84]. IGF-1 can act as a pro-carcinogenic factor, and IGFBP-3 is known to act as its regulator via its binding to IGF-1. In fact, IGF-1 and IGFBP-3 levels, as well as the IGF-1/IGFBP-3 ratio, are reported to be associated with cancer risk and tumor development [85,86,87]. Therefore, GTP might be a potentially useful supplement to decrease cancer risk and suppress the malignant aggressiveness of PC. Conversely, another study showed no significant change in serum PSA level or insulin-like growth factor axis, including IGF-1 and the IGF-1/IGFBP-3 ratio, in a randomized, double-blind, placebo-controlled trial of polyphenon E (800 mg/day for 3–6 weeks) in PC patients [88]. These two studies were performed in patients treated with radical prostatectomy. Therefore, no data on the changes in these biomarkers due to GTP were available in patients with advanced PC.

Conversely, several animal models, including a xenograft model using androgen-sensitive human PC cells and TRAMP mice, showed that reduced levels of IGF-I and increased levels of IGFBP-3 in serum were observed in mice orally administered GTP compared to those fed water [32,37,89]. However, in TRAMP mice, IGF-1 expression in epithelial cells of PC treated with GTP was significantly lower than that in those administered water alone [40,89]. Similarly, another study showed that the oral administration of GTP increased the IGF-1 level with a concomitant decrease in IGFBP-3 in the dorso-lateral prostate of TRAMP mice [47].

In addition to IGF-1 and IGFBP-3, EGCG can modulate the function of the IGF-1 receptor (IGF-1R) in human PC cell lines [33]. In short, EGCG inhibited IGF-1-induced PC growth by reducing the phosphorylation of IGF-1R in LNCaP and DU145 cells [33]. Furthermore, the oral administration of EGCG decreased IGF1R in ventral prostate tissues [40].

### 3.7. Nuclear Factor-κB

Nuclear factor (NK)-κB is a transcription factor involved in the regulation of a wide variety of biological responses, which play important roles in malignant aggressiveness in many types of cancers, including PC [66,90]. Anti-cancer effects, including cancer prevention caused by EGCG, were speculated to be mediated by a reduction in NK-κB activity in an in vivo study performed using an autochthonous mouse PC model [44,53]. In addition, oral infusion of GTP led to decreased NF-κB expression in athymic nude mice implanted with androgen-sensitive CWR22Rv1 [32]. Furthermore, the oral administration of GTP into an animal model led to changes in NK-κB expression and its regulated gene products. In short, when TRAMP mice were administered 0.1% GTP as drinking fluid three times a week for 32 weeks, the expression levels of phosphor-NK-κB, IKKα, IKKβ, receptor activator of NK-κB (RANK), NK-κB inducing kinase (NIK), and signal transducer and activator of transcription (STAT)-3 in the dorso-lateral prostate were significantly decreased compared to those in control mice [53]. Thus, NK-κB and NK-κB-associated components are generally accepted to play important roles in anti-cancer effects in PC because they are closely associated with the regulation of malignant behaviors, including tumor growth, apoptosis, angiogenesis, and invasion [91,92]. In addition, the crosstalk between NK-κB and other GTP-associated molecules such as AR, the IGF/IGF receptor axis, and COX-2 is known to be important for the regulation of malignant aggressiveness and outcome of PC [91,93,94]. Thus, NK-κB and the NK-κB-associated signaling system are recognized as a target of GTP to suppress malignant progression in PC.

### 3.8. Others

One theory is that inflammation of the prostate is closely associated with the increased risk of PC [95,96]. Increased plasma concentrations of various inflammatory chemokines such as interleukin (IL)-1, IL-6, and IL-8; interferon-γ; and tumor necrosis α were reported in patients with advanced PC and CRPC [97,98]. An in vitro study showed that EGCG suppressed the induction of the cytokines and chemokine genes IL-6, IL-8, CXCL-1, IP-10, CCL-5, and THG-β in LNCaP, DU145, and PC-3 cells and protected them from inflammation, which contributes to the tumor development of PC [99]. In addition, interestingly, these anti-inflammatory effects of EGCG were independent of AR and p53 status [99].

S100A4, also known as Mts1, is a member of the S100 calcium-binding protein family and has been associated with carcinogenesis and tumor development and progression in various types of cancers, including PC [100,101]. Conversely, E-cadherin is well known as an invasion suppressor because it plays important roles in homophilic cell-cell adhesion and substrate attachment and subsequently in the suppression of cancer cell motility [102]. In fact, decreased expression and/or mutation of E-cadherin are positively associated with high stage and poor prognosis in several malignancies [103,104,105]. In addition, the pathological significance of E-cadherin was detected in TRAMP mice [106]. Furthermore, interestingly, S100A4 expression is reported to be inversely correlated with E-cadherin expression in various types of malignancies [107,108]. With regard to the changes in S100A4 and E-cadherin expression following GTP administration, when TRAMP mice were supplied orally with 0.1% GTP solution three times a week for 24 weeks, tumor growth and progression were remarkably suppressed, which was associated with a reduction in S100A4 and restoration of E-cadherin in dorso-lateral prostate tissues [109]. Unfortunately, no additional data are available on the changes in S100A4 and E-cadherin following GTP administration in PC. Further studies are warranted to elucidate such issues.

Cyclooxygenase (COX)-2 is well known to be closely associated with carcinogenesis and malignant aggressiveness in various types of malignancies, including PC [110,111,112]. EGCG has been reported to inhibit COX-2 without affecting COX-1 expression at both the mRNA and protein levels in LNCaP and PC-3 human prostate carcinoma cells [93]. Conversely, in an animal model (TRAMP mice), EGCG also decreased COX-2 expression in ventral prostate tissues [40].

miRNAs are 18- to 24-nt small regulatory RNAs that repress target gene expression, and deregulation of miRNAs has been found in PC [113]. Little information is available on the influence of GTP on miRNAs in malignancies. However, in tumor xenograft tissues, the level of miRNA-21, which promotes hormone-dependent and hormone-refractory PC growth, in tissues isolated from animals treated with EGCG (1 mg/day, three times per week for 6 weeks) was 1.5-fold downregulated compared with that in controls [35,114]. In contrast, the same study also showed a >6-fold increase in the levels of miR-330, which induced apoptosis in PC cells in the tumor tissues of animals treated with EGCG, unlike in those of vehicle-treated controls [35,115]. Unfortunately, information on the changes in miRNA by GTP is lacking. We suggest further detailed studies are required to address this issue.

The vascular endothelial growth factor (VEGF) family is considered to play an important role in the carcinogenesis and progression of PC [116,117]. An animal model implanted with androgen-sensitive human PC cells (CWR22Rv1 cells) showed that the oral intake of GTP decreased the expression levels of VEGF in tumors [39]. A similar result was reported in the dorso-lateral prostate of TRAMP mice [46]. VEGF is the best-described angiogenetic factor under pathological condition, and its expression was closely associated with tumor progression and survival in PC [118,119]. In addition, interestingly, VEGF is reported to be associated with androgen-regulated angiogenesis in PC [120]. Conversely, in patients with PC, the serum levels of VEGF were significantly reduced after treatment with polyphenon E, which contained 800 mg of EGCG and lesser amounts of EC, EGC, and ECG (a total of 1.3 g of tea polyphenols) [84]. From these data, GTP is speculated to suppress carcinogenesis and tumor development via the regulation of angiogenesis in PC.

A nutrient mixture containing lysine, proline, ascorbic acid, and green tea extract has been reported to decrease urokinase plasminogen activator (uPA) expression in DU145 and PC-3 cells [79]. In addition, the level of uPA in the dorso-lateral prostate was decreased by oral administration of GTP in TRAMP mice [47]. The activation of uPA plays important roles in carcinogenesis and cancer cell invasion [121,122]. The expression of uPA has been reported to be positively associated with the prognosis and outcome of various types of cancer [123,124]. Therefore, we suggest that further studies need to be conducted to understand the pathological roles of uPA regulated by GTP.

Inducible nitric oxide synthase (iNOS) has been reported to influence malignant behaviors, including cell proliferation and invasion, by inducing the accumulation of mutations in p53, enrichment of basal-like gene signature, transactivation of the epidermal growth factor receptor, activation of NF-κB, and increased secretion of pro-inflammatory cytokines in malignancies, including PC [125,126]. One study showed that the oral administration of EGCG decreased iNOS activity in the ventral prostate of TRAMP mice [40]. Another study showed that minichromosome maintenance protein 7 (MCM7), which mediated tumor growth and cancer cell invasion of PC, was significantly suppressed by oral administration of GTCs in TRAMP mice [127]. A summary of molecular changes following GTP administration in animal experiments is shown in Table 3.

Nonetheless, to identify the anti-cancer effects and pathological roles at the molecular level of GTPs, we must pay special attention to the difference in their concentrations between in vivo and in vitro studies. Briefly, although the half maximal inhibitory concentration of tumor growth in PC was reported to range from 40 to 80 μM, it depended on the types of cell lines used, length of treatment, and method of administration [128]. In addition, a study suggested that EGCG plasma concentration was approximately 1/50–1/100 of the concentration used in vitro [129]. When evaluating the anti-cancer effects and biological activities of GTPs, the difference in concentration between in vivo and in vitro studies should be noted. In particular, concentrations of GTPs should be especially noted in discussions of the safety and adverse events caused by green tea, as these factors may be associated [130,131].

## 4. Conclusions

Taken together, green tea seems promising in terms of chemopreventive and therapeutic effects against PC, but there is no established evidence. A summary of currently published data obtained by in vitro studies, animal experiments, and clinical trials is presented in this review. In addition to meticulously selecting validated biomarkers and study endpoints, future PC chemoprevention and treatment trials should ideally enroll larger cohorts of men at higher risk for this disease, perhaps with duration of interventions beyond one year. Furthermore, although many cancer-related molecules have been reported to be associated with carcinogenesis, tumor development, and prognosis of PC, detailed mechanisms of their anti-cancer effects and regulation are lacking. Further, other components of green tea might play important roles in the anti-cancer effects. For example, flavonoids from green tea were reported to inhibit carcinogenesis, tumor growth, and metastasis in several cancers [132,133]. Moreover, quercetin was recognized as one of the green tea flavonoids, and it is reported to induce apoptosis in PC xenograft model injected androgen-sensitive LAPC-4 cells. Thus, green tea contains various substances that contribute to its beneficial effects. Moreover, there is a possibility that various other cancer-related molecules including 67kDa laminin receptor, a receptor of EGCG, are associated with malignant behaviors in malignancies [134]. Furthermore, we should note that the bioavailability of GTPs depends on both the concentration and reaction time. In the future, a more detailed understanding of the in vivo and in vitro pathological roles of green tea are necessary to elucidate the mechanisms of its anti-cancer effects, the regulation of which is highly complex. On the other hand, there is little information on the anti-cancer effects of EGCG alone in clinical trials, although such efficacy was reported in in vivo studies and animal models [34,36,40]. Therefore, appraising the usefulness and limitations of green tea-based prevention and treatment of PC requires further large-scale studies of sufficient quality and stringency.

## Figures and Tables

**Table 1 molecules-24-00193-t001:** Characteristics of studies on green tea consumption and prostate cancer risk (case-control studies).

Population, Region	Highest Consumption, RR (95% CI)	Year [Ref]
Japanese, Japan	≥10 cups/day, 0.67 (0.27, 1.64)	2004 [19]
Chinese, China	>3 cups/day, 0.27 (0.15, 0.48)	2004 [7]
Chinese, China	Unknown, 0.59 (0.40, 0.87)	2014 [20]

RR, relative risk; CI, confidence interval; Ref, reference.

**Table 2 molecules-24-00193-t002:** Characteristics of studies on green tea consumption and prostate cancer risk (cohort study).

Population, Region	Highest Consumption, RR (95% CI)	Year [Ref]
Japanese, USA	Unknown, 1.47 (0.99, 2.19)	1989 [21]
Japanese, Japan	>5 times/day, 1.29 (0.84, 1.98)	2004 [22]
Japanese, Japan	≥5 cups/day, 0.85 (0.50, 1.43)	2006 [23]
Japanese, Japan	≥5 cups/day, 0.90 (0.66, 1.23)	2008 [24]
Chinese, Singapore	≥2 cups/day, 0.95 (0.62, 1.45)	2012 [25]

RR; relative ratio, CI; confidence interval; Ref, reference.

**Table 3 molecules-24-00193-t003:** Changes in cancer-related molecules after the administration of green tea polyphenols.

Molecules	Type	Sample	Cell Line	Effect	Year [Ref]
Akt	TRAMP	Tissue		↓	2004 [47]; 2009 [89]
AR	TRAMP	Tissue		↓	2007 [40]
	Xenograft	Tissue	CW22Rv1	↓	2011 [35]
Bcl-2	Xenograft	Tissue	CW22Rv1	↓	2006 [39]; 2007 [32]
	TRAMP	Tissue		↓	2008 [53]
Bax	Xenograft	Tissue	CW22Rv1	↑	2006 [39]; 2007 [32]
	TRAMP	Tissue		↑	2008 [53]
COX-2	TRAMP	Tissue		↓	2007 [40]
E-cadherin	TRAMP	Tissue		↑	2005 [109]
Erk 1/2	TRAML	Tissue		↓	2004 [47]; 2007 [40]; 2009 [89]
IGF-1	TRAMP	Serum		↓	2001 [37]; 2009 [89]
	TRAMP	Tissue		↓	2004 [47]; 2007 [40]; 2009 [89]
	Xenograft	Serum	CW22Rv1	↓	2007 [32]
IGF-1R	TRAMP	Tissue		↓	2007 [40]
IGFBP-3	TRAMP	Serum		↑	2001 [37]; 2009 [89]
	TRAMP	Tissue		↑	2004 [47]
	Xenograft	Serum	CW22Rv1	↑	2007 [32]
IKKα and β	TRAMP	Tissue		↓	2008 [53]
iNOS	TRAMP	Tissue		↓	2007 [40]
MCM7	TRAMP	Tissues		↓	2007 [127]
miRNA-21	Xenograft	Tissue	CW22Rv1	↓	2011 [34]
miRNA-330	Xenograft	Tissues	CW22Rv1	↑	2011 [34]
NF-κB	Xenograft	Tissue	CW22Rv1	↓	2007 [32]
	TRAMP	Tissue		↓	2008 [53]
NIK	TRAMP	Tissue		↓	2008 [53]
PARP	Xenograft	Tissue	CW22Rv1	↑	2007 [32]
PI3K	TRAMP	Tissue		↓	2004 [47] 2009 [89]
PPARγ	Xenograft	Tissue	CW22Rv1	↓	2007 [32]
RANK	TRAMP	Tissue		↓	2008 [53]
STAT-3	TRAMP	Tissue		↓	2008 [53]
S100A4	TRAMP	Tissue		↓	2005 [109]
u-PA	TRAMP	Tissue		↓	2004 [47]
VEGF	Xenograft	Tissue	CW22Rv1	↓	2006 [39]
	TRAMP	Tissue		↓	2004 [47]

AR, androgen receptor; Bcl, B-cell lymphoma; Bax, bcl-2-associated X protein; COX, cyclooxygenase; IGF, insulin-like growth factor; IGFBP, IGF-binding protein; iNOS, inducible nitric oxide synthase; MCM7, minichromosome maintenance protein; NFκB, nuclear factor kappa B; PARP, poly-ADP ribose polymerase; PPAR, peroxisome proliferator-activated receptor; STAT, signal transduction and activator of transcription; u-PA, urokinase-type plasminogen activator; VEGF, vascular endothelial growth factor.

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
