# Peer review of "Anti-Cancer Effects of Green Tea Polyphenols Against Prostate Cancer"

_molecules, 2019, doi:10.3390/molecules24010193_

Round 1

Reviewer 1 Report

This review by Dr. Yasuyoshi Miyata and collegues, focus on the effects of green tea polyphenols on prostate cancer (PC).

The manuscript needs of a minor check for English language and a general reorganization. The font size is not always the same. The authors should check and improve these aspects.

In particular, in the introduction section the authors should describe the PC in totality, considering also tumor microenvironment and carcinoma associated fibroblasts (CAFs) that are emerging fields and could exalt the appealing of the review, citing also: 1)

Genetics and biology of prostate cancer. Genes Dev. 2018 Sep 1;32(17-18):1105-1140. doi: 10.1101/gad.315739.118. Review. PubMed PMID: 30181359; PubMed Central PMCID: PMC6120714.

and 2)

 Non-genomic androgen action regulates proliferative/migratory signaling in stromal cells. Front Endocrinol (Lausanne). 2015 Jan 19;5:225. doi: 10.3389/fendo.2014.00225. eCollection 2014. Review. PubMed PMID: 25646090; PubMed Central PMCID: PMC4298220.

They also should insert the concept of castration resistant prostate cancer (CRPC) of paragraph 3 in the introduction section.

The paragraph 2.2 is improved by the table, but in this table the authors should add information about the scientific value and significance of the experiments reported. It is not immediately understandable, infact, if these experiments are valid or not. 

In general, a table resuming the studies analyzed in every section could be appreciable. 

Please, specify ECGC in the text the first time. 

Pictures resuming in vivo studies and in vitro studies results could be appreaciable.

Strength:
Comprehensive review.
Updated proceedings in the field (please, improve only with the suggested considerations and references)

Limitations:
The authors should summarize or underline the novel findings presented.

Improved art work in the accompanying figures/graphs is required

Author Response

We thank the reviewer for evaluating our manuscript. We agree with your opinions, and your suggestions and advice have helped us greatly improve the manuscript. Our responses to your comments are provided below (page and line numbers in the revised version of the manuscript are indicated).

< Specific comments of reviewer 1 >

1. The manuscript needs of a minor check for English language and a general reorganization. The font size is not always the same. The authors should check and improve these aspects.

(Answer)

   Thank you for bringing this to our attention. We have ensured that the font size is consistent throughout the manuscript. Further, we have revised the manuscript for language and better presentation. In addition, the manuscript has been proofread by a native English-speaking editor from a professional English language editing service.

2. In the introduction section the authors should describe the PC in totality, considering also tumor microenvironment and carcinoma associated fibroblasts (CAFs) that are emerging fields and could exalt the appealing of the review, citing also: 1) Genetics and biology of prostate cancer. Genes Dev. 2018;32(17-18):1105-1140 and 2) Non-genomic androgen action regulates proliferative / migratory signaling in stromal cells. Front Endocrinol (Lausanne). 2015;5:225.

(Answer)

   Thank you for your valuable suggestion. In the revised version of the manuscript, we added the contents regarding the importance of tumor microenvironment and cancer cells in the part where we discuss the biological and pathological characteristics of prostate cancer. Furthermore, these 2 articles were cited to support our discussion (reference 1 and 2 in 1. Introduction; 1st. paragraph, lines 3–4).

).

3. They also should insert the concept of castration resistant prostate cancer (CRPC) of paragraph 3 in the introduction section.

(Answer)

   We are in agreement with your opinion. We moved some sentences from “3. Therapeutic effect for castration resistant prostate cancer” to “1. Introduction.” In addition, we modified each section and added relevant information to clarify the aim of this review (1. Introduction; 3rd. paragraph, lines 1–12).

4. The paragraph 2.2 is improved by the table, but in this table the authors should add information about the scientific value and significance of the experiments reported. It is not immediately understandable, in fact, if these experiments are valid or not. 

(Answer)

   We thought that Table 2 does not add value to the information presented in “2.2.2. Green tea catechin intake” section. Therefore, we deleted Table 2 from the revised manuscript. We believe that the deletion of this table does not affect the contents of this section.

5. In general, a table resuming the studies analyzed in every section could be appreciable. 

(Answer)

   We are in agreement with your view. According to your suggestion, we divided Table 1 into 2 tables: case control study (new Table 1) and prospective cohort study (new Table 2). We believe that such modification helps understand the contents of each section.

6. Please, specify ECGC in the text the first time. 

(Answer)

   We apologize for this oversight; we have expanded the abbreviated form of EGCG at the first mention in the revised manuscript (1. Introduction; 2nd. paragraph, line 5).

7. Pictures resuming in vivo studies and in vitro studies results could be appreciable.

(Answer)

   We are in agreement with your view. However, obtaining an image depicting the results of in vivo and in vitro studies seemed slightly difficult. Therefore, we attempted to clearly introduce the usefulness and limitations of GTP intake for PC patients. We believe that the findings of these studies are clearly presented in the revised version of the manuscript.

Strength:
Comprehensive review.
Updated proceedings in the field (please, improve only with the suggested considerations and references)

(Answer)

   Thank you for your constructive comments. As per your suggestion, we have revised and improved the presentation of the manuscript by including relevant citations. 

Limitations:
The authors should summarize or underline the novel findings presented.

Improved art work in the accompanying figures/graphs is required

(Answer)

   We are in agreement with your view. We have added new information and novel findings regarding the anti-cancer effects of green tea in prostate cancer. Unfortunately, as above mentioned, we could not add a new figure/graphic in the revised version of the manuscript. However, we believe that the revised version is suitably modified to clearly discuss the findings without requiring any visual presentation.

Reviewer 2 Report

The manuscript presented by Miyata et al. is aimed at describing the state of the art on the antitumor effect  of polyphenols extracted from green tea, with a special focus on their molecular mechanism of action. All this with a particular attention to prostate cancer.
Unfortunately the manuscript has many flaws. The arguments are presented in a confused, often partial, and not entirely logical way. There is confusion in the description of epidemiological studies versus interventional clinical trials. Citations are often wrong (the reference numbering  may also be wrong, sometimes), or confusing. Authors often fail to draw logical conclusions. In many cases is evident that they have not been able to critically analyze and comment on the aforementioned publications with due care. Many important experimental data present in the literature have not been considered by the Authors. In particular, the description of the molecular partners of catechins and their mechanism of action is not satisfactory. Some sentences are often too general, or difficult to understand. Also the use of language needs to be improved because of mistypings. The typographic layout of the manuscript is not coherent and reflects a certain superficiality in its drafting.
On the whole, this manuscript is not useful to readers of the topic in its present form. This is a pity, because the topic is hot and deserve high scientific attention.

Author Response

We thank the reviewer for evaluating our manuscript. We agree with your opinions, and your suggestions and advice have helped us greatly improve the manuscript. Our responses to your comments are provided below (page and line numbers in the revised version of the manuscript are indicated).

< Comments of reviewer 2 >

1. Unfortunately, the manuscript has many flaws. The arguments are presented in a confused, often partial, and not entirely logical way. There is confusion in the description of epidemiological studies versus interventional clinical trials.

(Answer)

   We apologize for the lack of logical flow of information in our manuscript. We have considerably revised the manuscript for better presentation and clarity. We have deleted parts that seemed less relevant and rearranged some parts for improving the flow of information. Moreover, as the findings on the anti-cancer effects of green tea or green tea polyphenols in androgen-independent prostate cancer cells varied across studies, and information on the clinical benefits and prolongation of survival periods by green tea consumption in patients with castration resistant prostate cancer (CRPC) was lacking, we deleted the section discussing the therapeutic effect for CRPC from the revised version of the manuscript.

2. Citations are often wrong (the reference numbering may also be wrong, sometimes), or confusing. Authors often fail to draw logical conclusions. In many cases is evident that they have not been able to critically analyze and comment on the aforementioned publications with due care.

(Answer)

   We apologize for not being able to clearly present our interpretations and for the missing citations. We have checked all the citations and ensured that they are numerically ordered throughout the manuscript. We also corrected the ordering of the citations at the relevant instances in the manuscript. Further, we have considerably modified the content presented in the review for better clarity and presentation.

3. Many important experimental data present in the literature have not been considered by the Authors. In particular, the description of the molecular partners of catechins and their mechanism of action is not satisfactory. Some sentences are often too general, or difficult to understand.

(Answer)

   We are in agreement with your view. We considerably modified the manuscript to clarify the molecular mechanisms of green tea and catechins. In addition, we presented detailed information regarding the aspects discussed in the review by including the findings of all relevant studies in this field (For examples; 3.1.2 In vivo studies: 2nd. paragraph, lines 11 – 12 and 3.2.2. In vivo studies: 1st. paragraph, lines 5 – 7).

We believe that the revised version of the manuscript is better presented and more understandable.

4. Also the use of language needs to be improved because of mistypings. The typographic layout of the manuscript is not coherent and reflects a certain superficiality in its drafting.

(Answer)

   We apologize for the oversights. We had the manuscript checked by a native English-speaking editor from a professional English language editing service. We have corrected all typographical errors in the revised manuscript.

5. On the whole, this manuscript is not useful to readers of the topic in its present form. This is a pity, because the topic is hot and deserve high scientific attention.

(Answer)

   As per your suggestion, we have considerably revised the manuscript to present our findings clearly. We believe that the revised version of the manuscript addresses all issues raised by you and the other reviewers and is comprehensible.

Reviewer 3 Report

DEAR COLLEAGUES

I really liked your work, but I think you should review some important aspects:

1. In many of the in vivo studies mentioned in the paper the effect of green tea intake is assessed. The content of catechins (GTCs) in the different tea extracts may be different, thus justifying the absence of response and / or the conflicting effects. Most probably, the composition of the infusions is mentioned in the original works, and I believe this should be reflected or commented in the review.

2. In addition to GTCs, green tea contains other polyphenols and / or substances with proven effects on tumor growth in vitro and / or in vivo. It should also be stated and discussed if other components of green tea may affect or be responsible for the beneficial effect attributed to the consumption of green tea.

3. In some sections of the work, the possible effect of green tea consumption in the prevention of prostate cancer is discussed, making reference to the prevention of malignancy and / or its therapeutic effects. I think the differences should be qualified.

4. In the section Cell cycle: what concentrations of EGCG are we talking about? Are these concentrations achievable in vivo? How could they affect healthy tissues?

5. When referring to androgenic receptors, do they refer to the response to T or DHT? The androgen-dependent prostate cancer is fundamentally sensitive to DHT, which explains why patients are treated preventatively with FINASTERIDE. It would be interesting to differentiate between effects of T or DHT. Are there studies on the effects of green tea or EGCG on the activity of 5a-reductase?

6. In section 4.8. Others:
    A) "EGCG was reported to suppress induction of the cytokines and chemokine genes IL-6, -8, CXCL-1, IP-10, CCL-5, and THG-β in LNCaP, DU145, and PC-3 cells, and protect them from inflammation, which contributes to the tumor development of PC". It is needed a clear distinction between in vivo and in vitro effects.

   B) In the case of effects tested in vitro, it would be advisable to comment if the concentrations of catechins (GTCs) are achievable in vivo.

    C) mts1 should be capitalized.

Author Response

We thank the reviewer for evaluating our manuscript. We agree with your opinions, and your suggestions and advice have helped us greatly improve the manuscript. Our responses to your comments are provided below (page and line numbers in the revised version of the manuscript are indicated).

< Comments of reviewer 3 >

1. In many of the in vivo studies mentioned in the paper the effect of green tea intake is assessed. The content of catechins (GTCs) in the different tea extracts may be different, thus justifying the absence of response and / or the conflicting effects. Most probably, the composition of the infusions is mentioned in the original works, and I believe this should be reflected or commented in the review.

(Answer)

   We are in agreement with your view. Based on your suggestion, we have added more detailed information on the composition of infusions at all relevant instances in the document. (For examples; 3.2.2. In vivo studies: 1st. paragraph, line 4, 5, and 10; 2nd. paragraph, line 4; 3.6 Insulin like growth factor: 2nd paragraph, line 3; 3.8 Others: 2nd. paragraph, line 10).

2. In addition to GTCs, green tea contains other polyphenols and / or substances with proven effects on tumor growth in vitro and / or in vivo. It should also be stated and discussed if other components of green tea may affect or be responsible for the beneficial effect attributed to the consumption of green tea.

(Answer)

   Thank you for bringing this to our attention. We agree that green tea includes various substances that have beneficial anti-cancer effects. For example, flavonoids from green tea are well known to inhibit carcinogenesis, tumor growth, and progression in various cancers. Based on your opinion, we have added this information and cited relevant references in the revised manuscript (reference [120 and 121] into 4. Conclusion” section (lines 8 -12).

3. In some sections of the work, the possible effect of green tea consumption in the prevention of prostate cancer is discussed, making reference to the prevention of malignancy and / or its therapeutic effects. I think the differences should be qualified.

(Answer)

   Thank you for bringing this to our attention. We have presented the preventive and therapeutic effects separately in the revised version of the manuscript.

4. In the section Cell cycle: what concentrations of EGCG are we talking about? Are these concentrations achievable in vivo? How could they affect healthy tissues?

(Answer)

   We have included information regarding the concentration and influence of these ECGC concentrations on non-tumoral tissues to highlight the clinical usefulness of green tea and green tea polyphenols in cancer patients. We have added relevant information and associated citations in the revised manuscript (reference [55] into “4.3. Cell cycle” section (1st. paragraph, lines 8 – 17).

5. When referring to androgenic receptors, do they refer to the response to T or DHT? The androgen-dependent prostate cancer is fundamentally sensitive to DHT, which explains why patients are treated preventatively with FINASTERIDE. It would be interesting to differentiate between effects of T or DHT. Are there studies on the effects of green tea or EGCG on the activity of 5a-reductase?

(Answer)

We found several studies on the relationship between green tea polyphenols and 5a-reductase in prostate cancer. In addition, in LNCaP cells, the relationship between sensitization of dihydrotestosterone for apoptosis and EGCG has been reported. Therefore, we presented these findings and included the relevant references in the revised manuscript. However, unfortunately, there was little information regarding this relationship in PC. Your questions are very interesting and important to understand the anti-cancer effects of green tea and green tea polyphenols in prostate cancer, and we have discussed these at relevant instances in the revised manuscript (3.4. Androgen receptor; 3rd. paragraph, lines 1 – 7).

6. In section 4.8. Others:
    A) "EGCG was reported to suppress induction of the cytokines and chemokine genes IL-6, -8, CXCL-1, IP-10, CCL-5, and THG-β in LNCaP, DU145, and PC-3 cells, and protect them from inflammation, which contributes to the tumor development of PC". It is needed a clear distinction between in vivo and in vitro effects.

(Answer)

   We are in agreement with your view. We have specified that the anti-inflammatory activities via the regulation of cytokines and growth factor by EGCG in PC were investigated in vitro (3.8. Others” section; 1st. paragraph, line 4).

   B) In the case of effects tested in vitro, it would be advisable to comment if the concentrations of catechins (GTCs) are achievable in vivo.

(Answer)

   We also believe that understanding the difference in concentrations between in vitro and in vivo studies is important to discuss the clinical benefit and inhibitory mechanisms of green tea catechins in PC. We have included relevant information and citations in the revised manuscript (reference [118] and [119] about it into last paragraph, lines 1–6 of “3. Results” section).

   C) mts1 should be capitalized.

(Answer)

   As per your suggestion, we have capitalized “mts1” in the revised manuscript (3.8. Others; 2nd paragraph, line 1).

Round 2

Reviewer 2 Report

The new version of the manuscript by Miyata et al. is considerably improved. Nevertheless, other flaws need to be addressed by the Authors.

1.       there is no mention of the discovery of other important mediators of the molecular action of EGCG or catechins.  For instance, the 67kDa laminin receptor (Tachibana et al., 2004), or MCM7 (McCarty et al., 2007);

2.       the reader may believe that green tea catechins or EGCG may only be inductors of apoptotic death: at difference, these compounds have been shown to be able to cause other kinds of cell death, including  anoikis or necroptosis (as an example, see Rizzi et al., 2014). In addition, also autophagy is affected by EGCG.  

3.       the study by Kumar et al. (ref. 6) has not been clearly understood and cited by the Authors. Despite what is affirmed in the title and in the summary (…a key paper must be read entirely and explored in detail…), it is easy to show that if you leave on a side ASAPs and analyze the result of the trial only considering the HG-PIN cases, it was a success instead. The progression to prostate cancer was significantly less in HG-PIN patients treated with GTCs vs those receiving placebo, thus fully confirming the result by Bettuzzi et al. (ref. 5). The mistake done by Kumar et al. was to recruit also patients bearing ASAP, an histological entity (not a lesion, not a disease…) which has nothing to do with prostate cancer, as everybody knows in the field. This mistake was clearly made to increase the final number of recruited patients, but in the end this was an artifact, which negatively affected the final result;

4.       there are several useless repetition in the text, such as lines 181-186, which do not add anything to the discussion;

5.       it is not clear what is the rational for using the abbreviation GTCs or GTPs;

6.       page 7, line 271: can you help readers with a little more on clusterin? Briefly, why induction of clusterin is related to suppression of prostate carcinogenesis? Maybe a review, or a significant experimental work, may help. Just a suggestion: Sala et al., Adv Cancer Res 2009; Bettuzzi et al. Oncogene 2009.

7.       page 8, lines 347-349: while you are discussing the possible effect of a mixture of green tea catechins on the malignant behavior of prostate cancer cells, the question for you is the following: are you aware of clinical trials in which EGCG alone was found effective? If not, what is the conclusion we may draw from these in vivo data…?   

8.       page 9, line 402: …associated…? Is it the right word…?

Author Response

Responses to reviewer comments

Reviewer 2.

We thank the reviewer for evaluating our manuscript. We agree with your opinions, and your suggestions and advice have helped us to greatly improve the manuscript. Our responses to your comments are provided below (page and line numbers in the revised version of the manuscript are indicated).

< Specific comments of reviewer 2 >

1. There is no mention of the discovery of other important mediators of the molecular action of EGCG or catechins. For instance, the 67kDa laminin receptor (Tachibana et al., 2004), or MCM7 (McCarty et al., 2007);

(Answer)

Thank you for important suggestions. In the revised version of the manuscript, we introduced the 67kDa laminin receptor (section 4. Conclusion, lines 15–17) and MCM7 (section 3.8. Others,: 7th. Paragraph, lines 6–8), and we included these two references as follows:

127. McCarthy, S.; Caporali, A.; Enkemann, S.; Scaltriti, M.; Eschrich, S.; Davalli, P.; Corti, A.; Lee, A.; Sung, J.; Yeatman, T.J.; Bettuzzi, S. Green tea catechins suppress the DNA synthesis marker MCM7 in the TRAMP model of prostate cancer. Mol Oncol. 2007, 1, 196–204. doi: 10.1016/j.molonc.2007.05.007

132. Tachibana, H.; Koga, K.; Fujimura, Y.; Yamada, K. A receptor for green tea polyphenol EGCG. Nat Struct Mol Biol. 2004, 11, 380–381. doi: 10.1038/nsmb743

2. The reader may believe that green tea catechins or EGCG may only be inductors of apoptotic death: at difference, these compounds have been shown to be able to cause other kinds of cell death, including anoikis or necroptosis (as an example, see Rizzi et al., 2014). In addition, also autophagy is affected by EGCG.  

(Answer)

Thank you for your valuable opinion. According to your suggestion, we modified section 3.2. Apoptosis. In short, we changed “3.2. Apoptosis” to “3.2. Cell death,” and added a new sub-section 3.2.3. Other kinds of cell death.

In this section, we introduced the relationships between GTPs including EGCG and anoikis or autophagy, and emphasize the importance of understanding such GTP-induced cell death in PC described in the following manuscripts.

54. Rizzi, F.; Naponelli, V.; Silva, A.; Modernelli, A.; Ramazzina, I.; Bonacini, M.; Tardito, S.; Gatti, R.; Uggeri, J.; Bettuzzi, S. Polyphenon E(R), a standardized green tea extract, induces endoplasmic reticulum stress, leading to death of immortalized PNT1a cells by anoikis and tumorigenic PC3 by necroptosis. Carcinogenesis. 2014, 35, 828–839. doi: 10.1093/carcin/bgt481

55. Wen, S.; Niu, Y.; Lee, S.O.; Chang, C. Androgen receptor (AR) positive vs negative roles in prostate cancer cell deaths including apoptosis, anoikis, entosis, necrosis and autophagic cell death. Cancer Treat Rev. 2014, 40, 31–40. doi: 10.1016/j.ctrv.2013.07.008

56. Modernelli, A.; Naponelli, V.; Giovanna Troglio, M.; Bonacini, M.; Ramazzina, I.; Bettuzzi, S.; Rizzi, F. EGCG antagonizes Bortezomib cytotoxicity in prostate cancer cells by an autophagic mechanism. Sci Rep. 2015, 5, 15270. doi: 10.1038/srep15270

57. Naponelli, V.; Modernelli, A.; Bettuzzi, S.; Rizzi, F. Roles of autophagy induced by natural compounds in prostate cancer. Biomed Res Int. 2015, 2015, 121826. doi: 10.1155/2015/121826

58. Naponelli, V.; Ramazzina, I.; Lenzi, C.; Bettuzzi, S.; Rizzi, F. Green Tea Catechins for Prostate Cancer Prevention: Present Achievements and Future Challenges. Antioxidants (Basel). 2017, 6, E26. doi: 10.3390/antiox6020026

3. The study by Kumar et al. (ref. 6) has not been clearly understood and cited by the Authors. Despite what is affirmed in the title and in the summary (…a key paper must be read entirely and explored in detail…), it is easy to show that if you leave on a side ASAPs and analyze the result of the trial only considering the HG-PIN cases, it was a success instead. The progression to prostate cancer was significantly less in HG-PIN patients treated with GTCs vs those receiving placebo, thus fully confirming the result by Bettuzzi et al. (ref. 5). The mistake done by Kumar et al. was to recruit also patients bearing ASAP, an histological entity (not a lesion, not a disease…) which has nothing to do with prostate cancer, as everybody knows in the field. This mistake was clearly made to increase the final number of recruited patients, but in the end this was an artifact, which negatively affected the final result;

(Answer)

Thank you for important comment. In the revised version of the manuscript, we deleted such sentences from section 2.2.2. Green tea catechin intake to avoid confusion.

4. There are several useless repetition in the text, such as lines 181-186, which do not add anything to the discussion;

(Answer)

We agree with your opinion. Therefore, we checked all sentences and deleted repetition in the text; for example, lines 134–135, 181–186, and 394–395 in text-R1.

5. It is not clear what is the rational for using the abbreviation GTCs or GTPs;

(Answer)

As you know, GTCs are recognized as a part of GTPs. On the other hand, realistically, both of them are used in the manuscript regarding biological activities and anti-cancer effects of green tea. In the revised version of the manuscript, we added the comment on the rational between GTCs and GTPs into section 1. Introduction (2nd. paragraph, lines 3 – 4).

6. Page 7, line 271: can you help readers with a little more on clusterin? Briefly, why induction of clusterin is related to suppression of prostate carcinogenesis? Maybe a review, or a significant experimental work, may help. Just a suggestion: Sala et al., Adv Cancer Res 2009; Bettuzzi et al. Oncogene 2009.

(Answer)

Thank you for your suggestion. We added the sentence on pathological roles of clusterin in PC (section 3.2.2. Apoptosis - in vivo studies, 1st. paragraph, lines 8–9) and cited the following two new references based on your suggestion.

49. Bettuzzi, S.; Davalli, P.; Davoli, S.; Chayka, O.; Rizzi, F.; Belloni, L.; Pellacani, D.; Fregni, G.; Astancolle, S.; Fassan, M.; Corti, A.; Baffa, R.; Sala, A. Genetic inactivation of ApoJ/clusterin: effects on prostate tumourigenesis and metastatic spread. Oncogene. 2009, 28, 4344–4452. doi: 10.1038/onc.2009.286

50. Sala, A.; Bettuzzi, S.; Pucci, S.; Chayka, O.; Dews, M.; Thomas-Tikhonenko, A. Regulation of CLU gene expression by oncogenes and epigenetic factors implications for tumorigenesis. Adv Cancer Res. 2009, 105, 115–132. doi: 10.1016/S0065-230X(09)05007-6

7. Page 8, lines 347-349: while you are discussing the possible effect of a mixture of green tea catechins on the malignant behavior of prostate cancer cells, the question for you is the following: are you aware of clinical trials in which EGCG alone was found effective? If not, what is the conclusion we may draw from these in vivo data…?   

(Answer)

Thank you for important question. To our knowledge, there is no clinical trial that show anti-cancer effects of EGCG alone in patients with PC. Significant effects were reported in just in vitro studies and mouse models. Therefore, we added such information into 4. Conclusions: lines 16 – 18.

8. Page 9, line 402: …associated…? Is it the right word…?

(Answer)

   We are sorry for the misunderstanding. We changed “associated” to “can modulate the function of” (section 3.6. Insulin-like growth factors, 3rd. paragraph, line 1).

Reviewer 3 Report

1. In many of the in vivo studies mentioned in the paper the effect of green tea intake is assessed. The content of catechins (GTCs) in the different tea extracts may be different, thus justifying the absence of response and / or the conflicting effects. Most probably, the composition of the infusions is mentioned in the original works, and I believe this should be reflected or commented in the review.

(Answer)

We are in agreement with your view. Based on your suggestion, we have added more detailed information on the composition of infusions at all relevant instances in the document. (For examples; 3.2.2. In vivo studies: 1st. paragraph, line 4, 5, and 10; 2nd. paragraph, line 4; 3.6 Insulin like growth factor: 2nd paragraph, line 3; 3.8 Others: 2nd. paragraph, line 10).

This additional information is frankly insufficient and does not clarify my concerns.

2. In addition to GTCs, green tea contains other polyphenols and / or substances with proven effects on tumor growth in vitro and / or in vivo. It should also be stated and discussed if other components of green tea may affect or be responsible for the beneficial effect attributed to the consumption of green tea.

(Answer)

Thank you for bringing this to our attention. We agree that green tea includes various substances that have beneficial anti-cancer effects. For example, flavonoids from green tea are well known to inhibit carcinogenesis, tumor growth, and progression in various cancers. Based on your opinion, we have added this information and cited relevant references in the revised manuscript (reference [120 and 121] into 4. Conclusion” section (lines 8 -12).

Again, based on this information (which does not include any discussion) the reader has not enough information to reach a conclusion on what is more critcal in the antitumor effects elicited by green tea extracts.

3. In some sections of the work, the possible effect of green tea consumption in the prevention of prostate cancer is discussed, making reference to the prevention of malignancy and / or its therapeutic effects. I think the differences should be qualified.

(Answer)

Thank you for bringing this to our attention. We have presented the preventive and therapeutic effects separately in the revised version of the manuscript.

In my opinion the authors failed in clearly presenting the effects of GT in either prevention and therapy.

4. In the section Cell cycle: what concentrations of EGCG are we talking about? Are these concentrations achievable in vivo? How could they affect healthy tissues?

(Answer)

We have included information regarding the concentration and influence of these ECGC concentrations on non-tumoral tissues to highlight the clinical usefulness of green tea and green tea polyphenols in cancer patients. We have added relevant information and associated citations in the revised manuscript (reference [55] into 4.3. Cell cycle” section (1st. paragraph, lines 8 – 17).

Cell cycle is actually 3.3. in the revised version. Like this, other coments have been made mentioning the first version, which makes this revision very complicated.

The main flaw here is the lack on any mention to in vivo bioavailability. This is essential to draw reliable conclusions when trying to correlate in vitro an in vivo effects.

5. When referring to androgenic receptors, do they refer to the response to T or DHT? The androgen-dependent prostate cancer is fundamentally sensitive to DHT, which explains why patients are treated preventatively with FINASTERIDE. It would be interesting to differentiate between effects of T or DHT. Are there studies on the effects of green tea or EGCG on the activity of 5a-reductase?

(Answer)

We found several studies on the relationship between green tea polyphenols and 5a-reductase in prostate cancer. In addition, in LNCaP cells, the relationship between sensitization of dihydrotestosterone for apoptosis and EGCG has been reported. Therefore, we presented these findings and included the relevant references in the revised manuscript. However, unfortunately, there was little information regarding this relationship in PC. Your questions are very interesting and important to understand the anti-cancer effects of green tea and green tea polyphenols in prostate cancer, and we have discussed these at relevant instances in the revised manuscript (3.4. Androgen receptor; 3rd. paragraph, lines 1 – 7).

The wording is incorrect and incomplete.

6. In section 4.8. Others:
A) "EGCG was reported to suppress induction of the cytokines and chemokine genes IL-6, -8, CXCL-1, IP-10, CCL-5, and THG-β in LNCaP, DU145, and PC-3 cells, and protect them from inflammation, which contributes to the tumor development of PC". It is needed a clear distinction between in vivo and in vitro effects.

(Answer)

We are in agreement with your view. We have specified that the anti-inflammatory activities via the regulation of cytokines and growth factor by EGCG in PC were investigated in vitro (3.8. Others” section; 1st. paragraph, line 4).

Again we face here the same problem as above. A main problem affecting the entire field of polyphenols and health is the lack of correlation between in vitro and in vivo effects. This is caused by ignoring that bioavailability is directly depending on two factors: concentration and time.  

B) In the case of effects tested in vitro, it would be advisable to comment if the concentrations of catechins (GTCs) are achievable in vivo.

(Answer)

We also believe that understanding the difference in concentrations between in vitro and in vivo studies is important to discuss the clinical benefit and inhibitory mechanisms of green tea catechins in PC. We have included relevant information and citations in the revised manuscript (reference [118] and [119] about it into last paragraph, lines 1–6 of “3. Results” section).

Unfortunately this information is insufficient and does not clarify this issue.

Author Response

Reviewer 3:

Comments to reviewer 3:

We thank the reviewer for evaluating our manuscript. We agree with your opinions, and your suggestions and advice have helped us to greatly improve the manuscript. Our responses to your comments are provided below (page and line numbers in the revised version of the manuscript are indicated).

Original Q1. In many of the in vivo studies mentioned in the paper the effect of green tea intake is assessed. The content of catechins (GTCs) in the different tea extracts may be different, thus justifying the absence of response and / or the conflicting effects. Most probably, the composition of the infusions is mentioned in the original works, and I believe this should be reflected or commented in the review.

(Answer for Original Q1)

We are in agreement with your view. Based on your suggestion, we have added more detailed information on the composition of infusions at all relevant instances in the document. (For examples; 3.2.2. In vivo studies: 1st. paragraph, line 4, 5, and 10; 2nd. paragraph, line 4; 3.6 Insulin like growth factor: 2nd paragraph, line 3; 3.8 Others: 2nd. paragraph, line 10).

Additional Q1. This additional information is frankly insufficient and does not clarify my concerns.

(Answer)

We understood your questions, and we are sorry our response was not sufficient. In the revised version of the manuscript, the composition of the infusion into the text (section 2.2.2. Green tea catechin intake, 1st. paragraph, line 8; section 3.1.2. In vivo studies, 1st. paragraph, line 2; section 3.2.2. Apoptosis – in vivo studies, 1st. paragraph, lines 3 and 6).

Original Q2. In addition to GTCs, green tea contains other polyphenols and / or substances with proven effects on tumor growth in vitro and / or in vivo. It should also be stated and discussed if other components of green tea may affect or be responsible for the beneficial effect attributed to the consumption of green tea.

(Answer for original Q2)

Thank you for bringing this to our attention. We agree that green tea includes various substances that have beneficial anti-cancer effects. For example, flavonoids from green tea are well known to inhibit carcinogenesis, tumor growth, and progression in various cancers. Based on your opinion, we have added this information and cited relevant references in the revised manuscript (reference [120 and 121] into 4. Conclusion” section (lines 8 -12).

Additional Q2: Again, based on this information (which does not include any discussion) the reader has not enough information to reach a conclusion on what is more critcal in the antitumor effects elicited by green tea extracts.

(Answer)

We agree with your opinion. Therefore, we added more detailed information and our comments into section 4. Conclusions. We also would like to provide more detailed information and conclusions on the mechanisms causing the anti-cancer effects of green tea in this review. However, our review already includes very much information on the anti-cancer effects at the clinical and molecular levels. We believe that the volume of this revised version of the manuscript is appropriate.

Original Q3. In some sections of the work, the possible effect of green tea consumption in the prevention of prostate cancer is discussed, making reference to the prevention of malignancy and / or its therapeutic effects. I think the differences should be qualified.

(Answer for original Q3)

Thank you for bringing this to our attention. We have presented the preventive and therapeutic effects separately in the revised version of the manuscript.

Additional Q3: In my opinion the authors failed in clearly presenting the effects of GT in either prevention and therapy.

(Answer)

   We agree with your opinion. Therefore, we have presented the preventive and therapeutic effects separately in section 3.1.2 In vivo studies. In short, in this section, we first describe the preventive effect of GTP in TRAMP mice, and then the therapeutic effects are reported.

Original 4. In the section Cell cycle: what concentrations of EGCG are we talking about? Are these concentrations achievable in vivo? How could they affect healthy tissues?

(Answer for original Q4)

We have included information regarding the concentration and influence of these ECGC concentrations on non-tumoral tissues to highlight the clinical usefulness of green tea and green tea polyphenols in cancer patients. We have added relevant information and associated citations in the revised manuscript (reference [55] into “4.3. Cell cycle” section (1st. paragraph, lines 817).

Additional Q4 – 1: Cell cycle is actually 3.3. in the revised version. Like this, other coments have been made mentioning the first version, which makes this revision very complicated. The main flaw here is the lack on any mention to in vivo bioavailability. This is essential to draw reliable conclusions when trying to correlate in vitro an in vivo effects.

(Answer)

We apologize for the typographical error. We understand what you say.

Unfortunately, there are few reports on the in vivo effects of GTPs on the cell cycle. Therefore, we added a statement regarding this to the text (section 3.3. Cell cycle, last sentence).

Original Q5. When referring to androgenic receptors, do they refer to the response to T or DHT? The androgen-dependent prostate cancer is fundamentally sensitive to DHT, which explains why patients are treated preventatively with FINASTERIDE. It would be interesting to differentiate between effects of T or DHT. Are there studies on the effects of green tea or EGCG on the activity of 5a-reductase?

 (Answer for original Q5)

We found several studies on the relationship between green tea polyphenols and 5a-reductase in prostate cancer. In addition, in LNCaP cells, the relationship between sensitization of dihydrotestosterone for apoptosis and EGCG has been reported. Therefore, we presented these findings and included the relevant references in the revised manuscript. However, unfortunately, there was little information regarding this relationship in PC. Your questions are very interesting and important to understand the anti-cancer effects of green tea and green tea polyphenols in prostate cancer, and we have discussed these at relevant instances in the revised manuscript (3.4. Androgen receptor; 3rd. paragraph, lines 1 – 7).

The wording is incorrect and incomplete.

(Answer)

In the previous revision (R1), we introduced the relationship between sensitization to dihydrotestosterone and EGCG in LNCaP cells (section 3.4. Androgen receptor, 4th paragraph). However, we agree with your opinion that it is incomplete. Therefore, we modified this section and presented additional information in the same paragraph. Furthermore, we added several sentences and the references below on the relationship between GTP and testosterone in animal models. We believe that these modifications will help the readers to understand the contents.

71. Figueiroa, M.S.; César Vieira, J.S.; Leite, D.S.; Filho, R.C.; Ferreira, F.; Gouveia, P.S.; Udrisar, D.P.; Wanderley, M.I. Green tea polyphenols inhibit testosterone production in rat Leydig cells. Asian J Androl. 2009, 11, 362–370. doi: 10.1038/aja.2009.2

72. Zhou, J.; Lei, Y.; Chen, J.; Zhou, X. Potential ameliorative effects of epigallocatechin‑3‑gallate against testosterone-induced benign prostatic hyperplasia and fibrosis in rats. Int Immunopharmacol. 2018, 64, 162–169. doi: 10.1016/j.intimp.2018.08.038

Original Q6. In section 4.8. Others:
A) "EGCG was reported to suppress induction of the cytokines and chemokine genes IL-6, -8, CXCL-1, IP-10, CCL-5, and THG-β in LNCaP, DU145, and PC-3 cells, and protect them from inflammation, which contributes to the tumor development of PC". It is needed a clear distinction between in vivo and in vitro effects.

(Answer for original Q6A)

We are in agreement with your view. We have specified that the anti-inflammatory activities via the regulation of cytokines and growth factor by EGCG in PC were investigated in vitro (3.8. Others” section; 1st. paragraph, line 4).

Again we face here the same problem as above. A main problem affecting the entire field of polyphenols and health is the lack of correlation between in vitro and in vivo effects. This is caused by ignoring that bioavailability is directly depending on two factors: concentration and time.  

(Answer)the

   We also think that the correlation between in vitro and in vivo effects is important for understanding the bioavailability of green tea in PC. However, unfortunately, there is no report on relationships between green tea intake and levels of cytokines or growth factor in vivo. We think that your question is important to discuss the biological activities of EGCG. Therefore, we added the comment about the importance of further large-scale studies into last sentences in 4. Conclusions section.

Original Q6B: In the case of effects tested in vitro, it would be advisable to comment if the concentrations of catechins (GTCs) are achievable in vivo.

(Answer for original Q6B)

We also believe that understanding the difference in concentrations between in vitro and in vivo studies is important to discuss the clinical benefit and inhibitory mechanisms of green tea catechins in PC. We have included relevant information and citations in the revised manuscript (reference [118] and [119] about it into last paragraph, lines 1–6 of “3. Results” section).

Additional Q6B: Unfortunately this information is insufficient and does not clarify this issue.

(Answer)

Unfortunately, we have no data about concentration of GTCs. In our previous revision (R1), the information about it was showed in the last paragraph of section 3.8. Others. However, based on your opinion, in the same paragraph we added the comment that the concentration difference between in vivo and in vitro studies should be noted when discussing the anti-cancer effects and biological activities of GTPs.